# Beyond the Surface: Nutritional Interventions Integrated with Diagnostic Imaging Tools to Target and Preserve Cartilage Integrity: A Narrative Review

**DOI:** 10.3390/biomedicines13030570

**Published:** 2025-02-24

**Authors:** Salvatore Lavalle, Rosa Scapaticci, Edoardo Masiello, Valerio Mario Salerno, Renato Cuocolo, Roberto Cannella, Matteo Botteghi, Alessandro Orro, Raoul Saggini, Sabrina Donati Zeppa, Alessia Bartolacci, Vilberto Stocchi, Giovanni Piccoli, Francesco Pegreffi

**Affiliations:** 1Department of Medicine and Surgery, Kore University of Enna, 94100 Enna, Italy; salvatore.lavalle@unikore.it (S.L.); valeriomario.salerno@unikore.it (V.M.S.); francesco.pegreffi@unikore.it (F.P.); 2Institute for the Electromagnetic Sensing of the Environment, National Research Council of Italy, 80124 Naples, Italy; scapaticci.r@irea.cnr.it; 3Department of Radiology, IRCCS San Raffaele Scientific Institute, 20132 Milan, Italy; 4Department of Medicine, Surgery, and Dentistry, University of Salerno, 84081 Baronissi, Italy; rcuocolo@unisa.it; 5Department of Biomedicine, Neuroscience and Advanced Diagnostics, University of Palermo, 90127 Palermo, Italy; 6Experimental Pathology Research Group, Department of Clinical and Molecular Sciences, Università Politecnica delle Marche, 60121 Ancona, Italy; matteo@botteghi.com; 7Medical Physics Activities Coordination Centre, Alma Mater Studiorum University of Bologna, 40126 Bologna, Italy; 8Institute of Biomedical Technologies CNR, Via Fratelli Cervi, 93, 20054 Segrate, Italy; alessandro.orro@cnr.it; 9Faculty of Psychology, eCampus University, 22060 Novedrate, Italy; raoulsaggini@gmail.com; 10Department of Biomolecular Sciences, University of Urbino Carlo Bo, 61029 Urbino, Italy; a.bartolacci2@campus.uniurb.it (A.B.); giovanni.piccoli@uniurb.it (G.P.); 11Department of Human Sciences for the Promotion of Quality of Life, University San Raffaele, 20132 Roma, Italy; vilberto.stocchi@uniroma5.it; 12Recovery and Functional Rehabilitation Unit, Ospedale Umberto I, 94100 Enna, Italy

**Keywords:** osteoarthritis, cartilage integrity, diagnostic tools, biochemical cartilage composition, biochemical markers, vitamin D and calcium, artificial intelligence, personalized strategies

## Abstract

This narrative review provides an overview of the various diagnostic tools used to assess cartilage health, with a focus on early detection, nutrition intervention, and management of osteoarthritis. Early detection of cartilage damage is crucial for effective patient management. Traditional diagnostic tools like radiography and conventional magnetic resonance imaging (MRI) sequences are more suited to detecting late-stage structural changes. This paper highlights advanced imaging techniques, including sodium MRI, T2 mapping, T1ρ imaging, and delayed gadolinium-enhanced MRI of cartilage, which provide valuable biochemical information about cartilage composition, particularly the glycosaminoglycan content and its potential links to nutrition-related factors influencing cartilage health. Cartilage degradation is often linked with inflammation and measurable via markers like CRP and IL-6 which, although not specific to cartilage breakdown, offer insights into the inflammation affecting cartilage. In addition to imaging techniques, biochemical markers, such as collagen breakdown products and aggrecan fragments, which reflect metabolic changes in cartilage, are discussed. Emerging tools like optical coherence tomography and hybrid positron emission tomography–magnetic resonance imaging (PET-MRI) are also explored, offering high-resolution imaging and combined metabolic and structural insights, respectively. Finally, wearable technology and biosensors for real-time monitoring of osteoarthritis progression, as well as the role of artificial intelligence in enhancing diagnostic accuracy through pattern recognition in imaging data are addressed. While these advanced diagnostic tools hold great potential for early detection and monitoring of osteoarthritis, challenges remain in clinical translation, including validation in larger populations and integration into existing clinical workflows and personalized treatment strategies for cartilage-related diseases.

## 1. Introduction

Articular cartilages are fundamental in joint function by providing a smooth surface for the articulation of bones and distributing mechanical loads to prevent damage to subchondral bone. However, the limited intrinsic repair capacity of cartilage poses a significant challenge when it is injured or degraded, often leading to conditions such as osteoarthritis (OA), which is a leading cause of chronic disability worldwide [1,2].

Early detection of cartilage damage is crucial for effective management and prevention of disease progression. Traditional diagnostic tools, such as radiography and morphological magnetic resonance imaging (MRI) sequences, primarily identify structural changes in cartilage which occur in later stages of disease [3]. However, these methods are limited in the assessment of early biochemical changes which precede morphological alterations. This has led to the development of advanced imaging techniques, such as sodium MRI, which can detect changes in cartilage glycosaminoglycan (GAG) content, a key component in maintaining cartilage integrity [4,5].

Moreover, novel biochemical markers and emerging molecular imaging techniques offer promising avenues for noninvasive assessment of cartilage health [6]. These tools not only enhance our ability to diagnose cartilage-related pathologies at earlier stages but may also provide valuable insights into the effectiveness of therapeutic interventions [7].

This narrative review aims to provide a comprehensive overview of the current diagnostic tools available for assessing cartilage integrity and health. It will discuss the anatomical and physiological underpinnings of cartilage, evaluate various imaging tech-niques and biochemical markers, explore emerging diagnostic tools, and consider the implications of these advancements for clinical practice.

## 2. Anatomy and Physiology of Cartilage

Articular cartilage is a specialized connective tissue found in synovial joints, where it serves as a load-bearing and friction-reducing surface [8]. The primary components of cartilage include water, collagen, proteoglycans, and chondrocytes, which together confer the tissue’s unique mechanical properties [9].

The extracellular matrix (ECM) of cartilage is predominantly composed of type II collagen fibers and proteoglycans, such as aggrecan, which are interwoven to form a dense, hydrated network. This network is crucial for resisting compressive forces, with the collagen fibers providing tensile strength and the proteoglycans contributing to the tissue’s ability to absorb water [10]. The negative fixed charge density (FCD) of proteoglycans, primarily due to the presence of GAG side chains, attracts positively charged ions like sodium, creating osmotic pressure which maintains the cartilage’s hydration and mechanical properties [11].

Chondrocytes, the only cell type present in cartilage, are responsible for maintaining the ECM by synthesizing and degrading its components. However, these cells are sparsely distributed within the matrix and have a limited capacity for proliferation and migration, which significantly hampers the tissue’s ability to repair itself after injury [12]. In adults, mechanical injury or degenerative changes often lead to irreversible cartilage damage, contributing to the development of OA [13].

Cartilage damage can occur due to degenerative changes, including the breakdown of collagen fibers and the loss of proteoglycans. This degradation compromises the structural integrity of the cartilage, leading to increased friction, reduced shock absorption, and ultimately the progression of OA [9]. The maintenance of cartilage health can be influenced by specific nutrients which support collagen synthesis and GAG preservation. Given the critical role of GAGs in maintaining cartilage integrity, their depletion is often an early indicator of cartilage degeneration [14]. Changes in GAG content are closely associated with the early stages of OA, even before morphological changes are detectable by traditional imaging techniques. Therefore, assessing the biochemical composition of cartilage, particularly GAG levels, has become a focal point in the development of advanced diagnostic tools [15].

Understanding the anatomy and physiology of cartilage is essential for the development and application of diagnostic techniques aimed at assessing cartilage health. These insights form the foundation for the use of imaging modalities and biochemical markers which can detect early degenerative changes, thereby enabling timely intervention and potentially slowing the progression of joint diseases.

## 3. Imaging Techniques for Assessing Cartilage Integrity

Imaging techniques are vital for diagnosing and managing diseases which affect cartilage. In clinical practice, traditional methods like X-rays and morphological MRI scans are commonly used to evaluate the structural integrity of cartilage. However, these approaches mainly detect late-stage alterations—such as narrowing of joint spaces and significant morphological changes—which are typical features of advanced OA (Table 1).

### 3.1. Magnetic Resonance Imaging (MRI)

MRI is widely regarded as the gold standard for the noninvasive assessment of cartilage due to its superior soft tissue contrast and ability to image cartilage [16]. Conventional MRI sequences, such as T2-weighted imaging, can provide detailed images of cartilage morphology, allowing for the evaluation of cartilage’s thickness, volume, and surface integrity using semi-quantitative scoring systems. However, standard imaging techniques are unable to detect microscopic cartilage lesions and early-stage degeneration [17]. To address this limitation, advanced MRI techniques, including T2 mapping, T1ρ imaging, and delayed gadolinium-enhanced MRI of cartilage (dGEMRIC) offer additional capabilities by providing additional information about cartilage’s composition [4,18].

### 3.2. T2 Mapping

T2 values can be calculated by acquiring multiple MRI images at different echo times (TEs). By plotting the signal intensity from these images against the echo times, an exponential decay curve is fitted. The T2 value is derived from the time constant of this decay curve, representing the rate at which the signal diminishes over time [19]. This technique is sensitive to the water content and collagen network integrity within cartilage. In cartilage, variations in T2 relaxation times are influenced by the amount of water and the condition of the proteoglycan-collagen matrix. Initial damage to the collagen matrix allows more water to enter, increasing permeability and causing stress within the matrix. This leads to further degeneration and loss of cartilage tissue, reflected by an elevated T2 signal. By assessing the spatial distribution of T2 relaxation times in articular cartilage, regions with increased or decreased water contents can be identified. Zhao et al. conducted a study to evaluate knee cartilage degeneration using quantitative MRI T2 mapping. The study included 66 patients and 28 healthy volunteers as a control group. Significant differences in T2 values were observed between patients and healthy individuals, particularly in the superficial cartilage of the patellofemoral joint. This finding demonstrates the utility of T2 mapping in the early detection of knee cartilage degeneration [20]. Alsayyad et al. demonstrated that incorporating a T2 mapping sequence into the standard 1.5 Tesla MRI protocol for assessing articular knee cartilage significantly improved the sensitivity for early OA detection, increasing from 73.3% to 96.7% [21]. A newer technique, ultrashort echo-time enhanced T2* (UTE-T2*) mapping, offers improved visualization of deep cartilage compared with standard T2 mapping. This method allows for calculation of the decay time between TEs and the signal intensity for each echo time within each voxel. Initial studies have shown that UTE-T2* mapping is effective in detecting cartilage degeneration and correlates well with morphological cartilage damage, indicating an altered cartilage structure [22,23]. These findings have been validated in vivo in OA patients compared with healthy controls [24].

### 3.3. T1ρ Imaging

T1ρ (T1 Rho) calculation is based on the spin-lattice relaxation in the rotating frame. It measures how spins relax when exposed to a spin-lock pulse, which is a radiofrequency (RF) pulse applied after the initial excitation pulse. The T1ρ value is calculated by fitting the decay of the MR signal as a function of the duration of the spin-lock pulse (TSL) [25]. The most commonly used sequences are FSE or TSE as well as balanced GRE sequences [26,27].

T1ρ imaging is particularly sensitive to the proteoglycan content within the cartilage matrix. By applying a TSL, T1ρ imaging enhances the contrast of tissues with different proteoglycan concentrations. In vitro studies confirmed a strong correlation (r^2^ = 0.987, slope = 0.95) between changes in the PG concentration and T1ρ, which has been shown to be a more sensitive and specific tool than T2 [28]. Furthermore, T1ρ is less influenced by the “magic angle effect” and results in a less distinct laminar appearance [29]. Studies have shown that individuals with early OA exhibit significantly elevated T1ρ relaxation times [30,31]. Prasad et al. showed that T2 and T1ρ measurements are potential indicators for predicting the progression of degenerative cartilage abnormalities in knee osteoarthritis. These values were significantly higher at the baseline in individuals who exhibited cartilage abnormality progression over a two-year period, suggesting their predictive value [32]. Similarly, Gallo et al. found that T1ρ and T2 relaxation parameters were associated with morphological cartilage degeneration in hip osteoarthritis over an 18-month period, further supporting their predictive value [33].

### 3.4. Delayed Gadolinium-Enhanced MRI of Cartilage (dGEMRIC)

One T1-mapping technique which quantitatively evaluates the GAG content of articular cartilage through intravascular or intraarticular injection of gadolinium contrast agents is dGEMRIC [34]. Since GAGs are negatively charged, they repel the negatively charged gadolinium ions. In regions with a reduced GAG content (a sign of cartilage degeneration), more gadolinium accumulates, resulting in shorter T1 relaxation times. Conversely, in healthy cartilage with higher GAG levels, less gadolinium is absorbed, leading to longer T1 relaxation times [35]. The dGEMRIC index, derived from post-contrast T1 relaxation times, offers a quantitative measure of the GAG concentration, aiding in the early detection and monitoring of cartilage degeneration. The concentration of Gd-DTPA^2^⁻ per voxel is calculated using a curve-fitting method from five different inversion times [36]. Palmer et al. demonstrated that baseline dGEMRIC measurements can predict the development of radiographic osteoarthritis, indicating its potential as an early diagnostic tool for assessing future cartilage degeneration [37]. In a study by Tjörnstrand et al., the findings in patients with AC injuries suggested that dGEMRIC has prognostic value for predicting the development of future knee OA [38]. A recent study evaluated the feasibility of using the dGEMRIC sequence to assess hip cartilage at 7T MRI following the intravascular administration of 0.2 mmol/kg Gd-DTPA^2^⁻. The high spatial resolution of 7T MRI, due to its increased signal-to-noise ratio (SNR), was highlighted in the study. However, the results showed no significant benefit from including pre-contrast T1 mapping [39]. One of the main drawbacks of the dGEMRIC sequence is the need for gadolinium-based contrast agents, which may have contraindications, especially for patients with kidney failure [40]. Additionally, the procedure requires a delay between the injection of the contrast agent and the MRI scan to allow adequate distribution of gadolinium within the cartilage, resulting in a longer imaging process. The sequence also has lower sensitivity in areas of thin cartilage, limiting its effectiveness in assessing certain joints. Furthermore, the dGEMRIC technique may be less accurate in advanced stages of osteoarthritis where cartilage is significantly degraded and the GAG content is already low [41].

### 3.5. Sodium MRI

Unlike conventional MRI techniques which rely on hydrogen protons (¹H-MRI), sodium MRI (^23^Na-MRI) directly measures the sodium ion concentration. Sodium ions are closely linked to the GAG content in cartilage, as GAGs attract sodium due to their negative charge. Therefore, ^23^Na-MRI can be used to indirectly measure the GAG concentration, which is critical for maintaining cartilage integrity. The sodium concentration within normal cartilage ECM can reach up to 300 mM and is proportional to the proteoglycan content. This relationship remains relatively unaffected by sodium levels or synovial inflammation [42]. One of the major drawbacks of ^23^Na-MRI is its low SNR, which is approximately 9% of the sensitivity of proton ^1^H-MRI. This limitation arises from the naturally low concentration of sodium in tissues, its short relaxation times, and the lower gyromagnetic ratio of sodium compared with hydrogen. As a result, the sodium MR signal is often difficult to detect. These challenges, however, can be mitigated by using high-field MRI scanners, such as 3T and ultrahigh-field 7T, which significantly improve the SNR and make the sodium signal more detectable. Despite these improvements, ^23^Na-MRI remains less commonly available and more technically demanding than conventional MRI scans [43,44]. Several in vitro and animal studies have shown that noninvasive sodium MRI can accurately measure the cartilage GAG content and detect its depletion in OA, resulting in a reduction in cartilage sodium levels. Although the sodium content is generally thought to decrease progressively in OA, a study by Newbould et al. found higher ^23^Na values in OA patients compared with an age-matched group of healthy controls using T1-weighted sodium MRI [45]. Additionally, sodium MRI has been found to be a less effective method for evaluating articular cartilage stiffness, though the findings remain a topic of ongoing debate [43].

### 3.6. Glycosaminoglycan Chemical Exchange Saturation Transfer (gagCEST)

In cartilage, gagCEST allows for the indirect detection and quantification of GAGs. The technical principle behind gagCEST involves selectively exciting the exchangeable protons in the hydroxyl groups of GAG molecules using a specific radiofrequency pulse. These protons undergo a chemical exchange with the abundant water protons in the tissue [46]. Once the protons in the GAGs are saturated, they are exchanged with water protons, effectively transferring this saturation to the water pool. This transfer causes a reduction in the magnetization of the water signal, which is then detected by MRI as a decrease in the signal intensity. The amount of signal reduction is proportional to the concentration of GAGs in the tissue, making it possible to map the spatial distribution of the GAG content in cartilage. This effect is typically quantified as an asymmetry in the magnetization transfer spectrum, referred to as MT asymmetry, or expressed as a percentage change in the water signal [47]. Areas with lower GAG contents display decreased MT and, consequently, lower asymmetry values [4]. Soellner et al. demonstrated that GAG-CEST imaging accurately reflects the GAG content and shows potential as a diagnostic tool for detecting early cartilage damage in the knee joint and differentiating between various International Cartilage Repair Society grades through noninvasive MRI, even in the early stages of its clinical use [48].

### 3.7. Hybrid PET-MRI

Positron emission tomography–magnetic resonance imaging (PET–MRI) is an advanced imaging technology which combines the detailed soft tissue images from MRI with the functional imaging capabilities of PET. Hybrid PET-MRI systems offer a comprehensive approach to imaging the entire joint, including both soft tissues and bone, which is essential for studying the complex pathophysiology of OA [49]. PET imaging, particularly with tracers like ^18^F-fluoride and ^18^F-FDG, provides metabolic insights into bone remodeling and inflammation, respectively [50]. These metabolic data can be combined with MRI’s structural details, allowing for a more complete assessment of the spatial interactions between tissues as OA progresses. The initial use of hybrid PET-MRI has demonstrated that metabolic activity differs across various subchondral bone pathologies detected through MRI. This integration of metabolic and structural imaging not only deepens our understanding of bone and joint changes but also enhances the interpretation of semi-quantitative MRI scoring systems, which are commonly employed in OA research [51]. Furthermore, ^18^F-fluoride PET has the potential to detect abnormal subchondral bone metabolic activity before structural changes are apparent in MRI, making it a promising tool for early OA diagnosis and intervention [52]. Despite its potential, there is a notable scarcity of data in the literature on the use of hybrid PET-MRI in cartilage assessment.

### 3.8. Ultrasound

Ultrasound (US) has emerged as a valuable tool in the diagnosis of OA, particularly over the past two decades, and it is now being recognized as an important complementary imaging modality due to its ability to assess soft tissue changes, bony surface contours, and inflammation related to OA [53]. The use of US in OA diagnostics has increased significantly, with a particular focus on knee and hand OA. Studies have shown that US is reliable for detecting osteophytes, joint effusion, synovitis, and meniscal protrusion, often outperforming CR in these areas [54,55]. For example, US has been shown to be particularly effective in evaluating cartilage damage, especially in the medial femoral condyle, and in detecting localized inflammation [56]. Although US cannot visualize intra-articular structures as effectively as MRI, it offers clear advantages in assessing superficial joint abnormalities [57].

In terms of accessibility, US is widely available, inexpensive, and can be used in primary healthcare settings, making it an attractive option for OA diagnostics. Its reproducibility and strong correlation with MRI findings further support its utility as a diagnostic tool [58,59]. US also holds prognostic value, as several studies have demonstrated its ability to predict OA progression based on findings like meniscal extrusion and synovitis [51,60]. These factors underscore US’s growing role in modern OA imaging, complementing traditional methods and offering a noninvasive, cost-effective approach to OA assessment.

### 3.9. Optical Coherence Tomography (OCT)

Optical coherence tomography (OCT) has shown significant promise in the diagnosis and management of OA, as outlined in the article. OCT is an infrared-based imaging technology capable of providing high-resolution, micron-scale images of joint tissues, including cartilage and bone [61]. Its ability to detect early changes in the tissue microstructure, which are often undetectable by other imaging modalities, makes it a valuable tool in OA research and diagnostics. OCT’s resolution, which is greater than 10 microns, allows for detailed visualization of cartilage health. It can detect early signs of OA, such as the breakdown of the collagen and glycosaminoglycan matrix, which are critical in maintaining cartilage’s structure and function. This early detection capability is crucial since OA progression involves irreversible damage, and early intervention could potentially slow or halt disease advancement [62]. OCT can image changes in cartilage even before significant cartilage thinning occurs, making it a proactive diagnostic tool. Despite its advantages, OCT is still relatively new technology in musculoskeletal disease, and the data supporting its widespread clinical application, especially in osteoarthritis, are still emerging. However, the initial results are promising, and ongoing clinical trials are exploring its potential further.

## 4. Biochemical Markers for Cartilage Health

In addition to imaging techniques, the assessment of biochemical markers provides a valuable approach to evaluating cartilage health [63]. These markers, which can be measured in body fluids such as blood, urine, and synovial fluid, reflect the metabolic processes occurring within the cartilage and can indicate the presence of degeneration or repair [64] (Table 2).

### 4.1. Collagen Breakdown Products

One of the most widely studied groups of biochemical markers in cartilage health is collagen breakdown products, including C-telopeptide of type II collagen (CTX-II) and collagen type II cleavage products (C2Cs). These markers are indicative of the degradation of type II collagen, a key structural component of cartilage, and their elevated levels are associated with cartilage degradation in OA [65]. A meta-analysis by Cheng et al. involving a total of 2856 participants demonstrated that urinary CTX-II levels were significantly elevated in the knee osteoarthritis (OA) group compared with the controls (SMD 0.82; 95% CI 0.41–1.24; *p* < 0.0001). This suggests that urinary CTX-II can serve as a potential biomarker for diagnosing and monitoring knee OA [66].

### 4.2. Aggrecan Fragments

Aggrecan is a major proteoglycan in cartilage, and its degradation results in the release of aggrecan fragments into the synovial fluid and bloodstream [67]. The measurement of these fragments serves as an indicator of cartilage breakdown, particularly in the early stages of OA, where aggrecan loss precedes collagen degradation [68,69]. A cross-sectional study by Larsson et al. found that the aggrecan fragment concentrations were significantly different across the OA group compared with the reference group (*p* < 0.001), suggesting that elevated levels of aggrecan ARGS fragments in synovial fluid are associated with this inflammatory process [70].

### 4.3. Inflammatory Markers

Cartilage degradation in OA is often accompanied by inflammation, which can be assessed through various inflammatory markers such as C-reactive protein (CRP) and interleukin-6 (IL-6). While these markers are not specific to cartilage degradation, their levels can provide insight into the inflammatory processes contributing to cartilage damage [71].

## 5. Challenges and Limitations

Despite the potential of biochemical markers to provide early indications of cartilage degeneration, several challenges limit their clinical utility. Variability in marker levels due to factors such as age, gender, and comorbid conditions can complicate interpretation. Moreover, the lack of standardized assays and the need for validation in large, diverse populations further limit the widespread adoption of these markers in clinical practice [72,73].

Nevertheless, the combination of biochemical markers with advanced imaging techniques holds promise for a more comprehensive assessment of cartilage health [74]. By integrating biochemical data with imaging findings, clinicians can obtain a more nuanced understanding of cartilage pathology, potentially leading to earlier and more targeted interventions.

## 6. Wearable Technology and Biosensors

The integration of wearable technology into cartilage diagnostics is an emerging field which aims to provide continuous monitoring of joint health in real time.

### 6.1. Wearable Sensors

Wearable sensors have shown significant potential in the assessment and monitoring of OA by providing real-time, objective data on joint movement, gait patterns, and physical activity. These sensors can be attached to various parts of the body, such as the knee, hip, or wrist, to track biomechanical parameters which are difficult to assess in a clinical setting. Studies have demonstrated the reliability of wearable sensors in capturing consistent data over time, making them useful for long-term OA management. Their ability to monitor daily activities and detect subtle changes in mobility and joint function offers a more personalized approach to tracking disease progression [75,76]. However, the accuracy and consistency of measurements can be influenced by sensor placement, calibration, and patient compliance [77]. While wearable sensors are generally reliable, further validation is needed to standardize their use in clinical practice for OA diagnosis and treatment monitoring.

### 6.2. Biosensors

Biosensors have emerged as promising tools for detecting and monitoring osteoarthritis OA by providing real-time, precise measurements of biochemical markers associated with the disease. These sensors can measure biomarkers such as cytokines, enzymes, and GAGs in biological fluids like synovial fluid, serum, or urine, offering noninvasive and dynamic insights into disease progression [78]. While biosensors show potential for early detection and personalized treatment strategies, their reliability in OA depends on several factors, including sensitivity, specificity, and reproducibility. Current biosensor technologies are being refined to improve their accuracy in detecting low-concentration biomarkers typical in the early stages of OA. Challenges such as sensor degradation over time, interference from other biological molecules, and the need for standardization across platforms can affect their consistency. Nevertheless, advances in biosensor design, including multi-analyte sensing and integration with digital health platforms, are enhancing their reliability, making them valuable for early diagnosis, monitoring, and treatment evaluation in OA [79].

## 7. 3D Bioprinting and Tissue Engineering

In the realm of cartilage repair and regeneration, 3D bioprinting and tissue engineering are pushing the boundaries of what is possible in cartilage diagnostics and treatment.

### 7.1. 3D Bioprinting

3D bioprinting allows for the precise fabrication of cartilage tissue constructs which can be used for both diagnostic and therapeutic purposes. These constructs can be designed to mimic the native cartilage structure and composition, providing a platform for testing new diagnostic techniques or evaluating the effectiveness of cartilage repair strategies [79,80,81].

Recent studies highlighted the clinical potential of 3D bioprinting in cartilage regeneration. For example, advanced bio-inks incorporating stem cells and growth factors have been successfully used to fabricate cartilage-like tissues with mechanical properties similar to native cartilage [82]. These bioengineered tissues have shown promising results in preclinical studies, particularly for osteochondral defect repair.

Additionally, bioprinted hydrogels embedded with mesenchymal stem cells (MSCs) have been applied in in vitro and in vivo models to enhance cartilage regeneration [83].

The use of scaffold-free 3D bioprinting has also been explored to create tissue constructs which mimic the zonal architecture of native cartilage, which is crucial for its functionality and durability.

In diagnostics, engineered cartilage tissues have been utilized as in vitro models for osteoarthritis drug screening. These models allow researchers to test potential therapeutics in a physiologically relevant environment, improving the translational potential of preclinical studies [84].

### 7.2. Tissue-Engineered Cartilage Models

Tissue-engineered cartilage models are also being developed as in vitro systems for studying cartilage pathology and screening potential therapeutic agents. These models offer a controlled environment where the effects of mechanical loading, biochemical factors, and therapeutic interventions on cartilage health can be systematically studied [85,86]. The insights gained from these models can inform the development of new diagnostic tools and treatment strategies.

## 8. Artificial Intelligence (AI)

Artificial intelligence (AI) is increasingly being integrated into cartilage diagnostics to enhance the accuracy and efficiency of imaging analysis. AI algorithms can be trained to recognize patterns in imaging data which are indicative of early cartilage degeneration, potentially identifying subtle changes that may be missed by human observers [87]. AI can also assist in the quantification of imaging biomarkers, such as the cartilage thickness or GAG content, providing more precise and reproducible measurements [88]. The systematic review and meta-analysis performed by Mohammadi et al. examined the performance of AI algorithms in detecting and classifying OA compared with clinicians. The study pooled data from 61 studies, focusing on diagnostic performance metrics such as sensitivity and specificity. A total of 27 studies with 91 contingency tables were included in the meta-analysis, evaluating both internal and external validation sets. The meta-analysis showed that the AI algorithms had a pooled sensitivity of 88% and specificity of 81% in internal validation compared with the clinicians’ sensitivity of 80% and specificity of 79%. In external validation, AI outperformed the clinicians, with a pooled sensitivity of 94% and specificity of 91%. This indicates that AI performs comparably to or even better than clinicians in detecting and classifying OA. The study also highlighted the potential role of AI as a diagnostic adjunct to radiologists [89].

Recent studies have demonstrated the effectiveness of deep learning (DL) and machine learning (ML) techniques in improving diagnostic accuracy for cartilage-related conditions. For instance, deep learning super resolution MRI has been shown to enhance image quality and improve diagnostic performance for detecting articular cartilage lesions compared with conventional MRI techniques [90]. This approach utilizes simultaneous multi-slice parallel imaging-accelerated DL models to reconstruct high-resolution images while maintaining diagnostic reliability.

Furthermore, convolutional neural networks (CNNs) have been successfully applied for automated classification of temporomandibular joint disorders using MRI scans, demonstrating their potential for analyzing cartilage structures in a clinical setting [91]. Similarly, studies have employed random forest (RF), support vector machine (SVM), and extreme gradient boosting (xGBoost) models to develop biomarker-based predictive tools for knee osteoarthritis [92]. These models integrate clinical and imaging data to improve diagnostic precision and risk stratification.

Additionally, artificial neural networks (ANNs) have been used to identify novel diagnostic and therapeutic targets for osteoarthritis, further demonstrating the role of AI in integrating bioinformatics and imaging data for personalized diagnostics [93]. The use of scaffold-free 3D bioprinting has also been explored to create tissue constructs which mimic the zonal architecture of native cartilage, which is crucial for its functionality and durability.

Despite the promising advancements of AI in musculoskeletal imaging, several key challenges hinder its widespread clinical adoption. One significant issue is data standardization, as AI models require large, high-quality, and diverse datasets to ensure generalizability and minimize bias. However, differences in imaging protocols, scanner types, and resolution across institutions complicate the development of universally applicable AI models [90].

Another critical challenge is model interpretability, particularly in deep learning applications. While convolutional neural networks (CNNs) and other deep learning models have demonstrated high accuracy in cartilage diagnostics, their “black box” nature makes it difficult for clinicians to understand how decisions are made, reducing trust in AI-assisted diagnoses [92].

Furthermore, regulatory concerns remain a major barrier. AI models used for medical imaging must comply with stringent regulatory frameworks, such as the FDA’s guidelines for AI-based medical devices and the European Medical Device Regulation (MDR). Ensuring AI algorithms meet these requirements while maintaining adaptability and continuous learning presents an ongoing challenge for AI developers [93].

To address these issues, future research should focus on developing explainable AI (XAI) approaches, improving data harmonization across imaging centers, and establishing standardized regulatory pathways for AI-driven diagnostics.

## 9. Clinical Translation and Challenges

While these emerging diagnostic tools hold great promise, several challenges must be addressed before they can be widely adopted in clinical practice. One of the primary challenges is the need for validation in large, diverse patient populations. Many of these techniques are still in the early stages of development and have been tested primarily in small-scale studies or preclinical models. Large-scale clinical trials are necessary to establish the reliability, specificity, and sensitivity of these tools in detecting cartilage degeneration and predicting disease progression [94].

Another challenge is the integration of these advanced tools into existing clinical workflows. The adoption of new diagnostic technologies often requires significant changes in clinical practice, including the need for specialized equipment, training, and changes to diagnostic protocols. Ensuring that these tools are accessible and cost-effective for widespread use is also a critical consideration [95]. Despite these challenges, the continued development and refinement of these emerging diagnostic tools offer the potential to revolutionize the field of cartilage diagnostics. By providing earlier and more accurate detection of cartilage degeneration, these tools could lead to improved outcomes for patients with OA and other cartilage-related conditions. The integration of these tools with existing diagnostic techniques, along with advances in personalized medicine and AI, will likely play a key role in the future of cartilage diagnostics [96].

## 10. Role of Diet in Improving Cartilage Health

Diet plays a crucial role in supporting joint health by helping reduce inflammation and combat oxidative stress. Anti-inflammatory diets are particularly effective in lowering markers such as C-reactive protein (CRP) and pro-inflammatory cytokines, both of which are known contributors to joint inflammation [97].

Obesity is a major risk factor for inflammation and knee osteoarthritis (OA), with its rising prevalence attributed to aging populations and increasing obesity rates [98]. Short-term weight loss can alleviate knee OA symptoms, with diet and exercise being preferred strategies [99]. Anti-inflammatory diets, particularly the phosphate very low-calorie ketogenic diet (VLCKD), have shown promise in reducing inflammation and improving bone structure by inhibiting pro-inflammatory cytokines and the NLRP3 inflammasome in rat models [100].

Originally developed for epilepsy, the low-carbohydrate diet is now applied to various conditions, including musculoskeletal disorders. The VLCKD has been proven to be effective for weight loss in individuals resistant to other diets and has been included in obesity management guidelines [101]. Preliminary evidence suggests that low-carbohydrate diets may alleviate pain in knee OA patients independent of weight loss. A pilot study by Ciaffi et al. (2024) demonstrated that a 20-week VLCKD significantly reduced weights and improved outcomes for the WOMAC Osteoarthritis Index, the EuroQol 5D (EQ-5D), and the 36-item Short Form Health Survey (SF-36) in women with obesity and symptomatic knee OA [98].

Another randomized, controlled pilot study evaluated the efficacy of low-carbohydrate diets (LCDs) and low-fat diets (LFDs) over 12 weeks. The LCD group showed reductions in pain intensity, unpleasantness, self-reported pain, oxidative stress, and adipokine leptin levels compared with the LFD and control groups, indicating a potential link between oxidative stress and functional pain [102].

## 11. Nutritional Supplementation in Improving Cartilage Health

Nutrients like vitamins C and E serve as antioxidants, effectively neutralizing reactive oxygen species (ROS) and protecting cartilage cells (or chondrocytes) from oxidative damage [103]. Intra-articular administration of vitamin C and Mg^2+^ alleviates joint destruction and pain in OA in mice [104]. Additionally, tryptophan, an essential amino acid, supports cartilage health by producing metabolites like kynurenic acid through the kynurenine pathway [105]. These metabolites play a significant role in alleviating inflammation and oxidative stress by helping to regulate the immune response and prevent cartilage degradation, especially in conditions like osteoarthritis. Vitamin C is especially important for the synthesis of collagen and proteoglycans, which are vital components of healthy cartilage. Consuming collagen-rich foods such as bone broth and gelatin or taking collagen supplements can further bolster cartilage health. Research indicates that collagen peptides can stimulate collagen production and enhance joint function in individuals with osteoarthritis [106]. The integration of therapeutic exercise with oral viscosupplementation of collagen peptides, vitamin C, sodium hyaluronate, manganese, and copper has demonstrated therapeutic efficacy in patients suffering from chronic lower back pain. This approach not only alleviates pain but also enhances the quality of life and improves the functionality of the lumbar spine [107].

Moreover, omega-3 fatty acids, found in fatty fish like salmon, mackerel, and sardines as well as in flaxseeds, chia seeds, and walnuts, possess strong anti-inflammatory properties which help ease joint inflammation [108]. Omega-3 fatty acids safeguard cartilage against acute injuries by diminishing the mechanical sensitivity of chondrocytes [109]. Meta-analysis carried out by Deng et al. reported that supplementation of n-3 PUFAs is effective for relieving pain and improving joint function in patients with OA [110]. Felson et al. (2024) did not observe any significant correlations between the levels of specific n-3 fatty acids, including eicosapentaenoic acid (EPA), or n-6 fatty acids, and the incidence of osteoarthritis (OA) indicates that further research is warranted to elucidate the role of n-3 fatty acids in the prevention and treatment of OA [111].

Protein is also essential for the repair and maintenance of cartilage, providing amino acids such as glycine, proline, and hydroxyproline, which are critical for collagen production [91]. Muscle function plays a pivotal role in managing pain and maintaining ambulatory function in knee osteoarthritis (OA), and a high plant protein and peptide supplementation can improve knee OA symptoms by enhancing muscle mass and strength. A randomized controlled study showed that 12-week supplementation with plant proteins and peptides significantly improved muscle mass, strength, and physical performance in elderly individuals with minor-to-mild knee OA and sarcopenia, as evidenced by health quality measures [112].

Supplementing with leucine-enriched essential amino acids (LEAAs) has been proven to be effective in promoting muscle injury recovery and stimulating muscle protein synthesis. Additionally, leucine-enriched protein supplementation has been shown to be both safe and efficacious in enhancing muscle density and overall quality of life [113,114].

Nutraceuticals, particularly dietary berries, have gained popularity for mitigating arthritis symptoms. Recent research has shown that strawberry supplementation can reduce pain and inflammatory markers in adults with knee OA [115]. Freeze-dried strawberries reduced TNF-α and lipid peroxidation products in obese adults with knee OA, warranting further investigation.

Dietary polyphenols, such as those found in blueberries, possess anti-inflammatory properties and may have anabolic effects on cartilage cells. A randomized double-blind trial indicated that daily consumption of 40 g of freeze-dried blueberry powder significantly reduced pain, stiffness, and difficulty in performing daily activities in individuals with symptomatic knee OA, with no significant changes in inflammatory markers [116].

Vitamin D is fundamental for the development and maintenance of strong bone structure. The active form of vitamin D aids in the absorption of calcium and phosphates through multiple pathways, enhancing bone mineralization and providing protection against bone deterioration and fractures [117,118]. Observational data suggests that vitamin D deficiency is associated with the initiation and progression of knee osteoarthritis (OA). Both elevated and reduced serum calcium levels, increased C-reactive protein (CRP), and vitamin D insufficiency are potential predictors of increased mortality risk in the osteoarthritis population [119]. However, the relationship between serum vitamin D levels and OA, as well as the efficacy of vitamin D supplementation in preventing knee OA, remains controversial. However, thanks to new administration methods like Filmtec^®^ technology, the orodis-persible film—a flexible, ultra-thin sheet (50–150 µm thick) which dissolves within seconds upon contact with saliva—offers a novel dosage form that ensures a precise and uniform concentration of the active ingredient, rapid bioavailability, and ease of intake under various conditions [120].

Calcium is another debatable nutrient. Recent evidence highlights the potential adverse effects of excessive calcium intake (particularly from supplements) on arterial calcification and cardiovascular disease (CVD) risks in older adults. Studies indicate that high calcium intake in free-living adults offers minimal or no benefits for bone mineral density (BMD) and fracture rates, suggesting that current calcium recommendations for adults may be overly high. Consequently, while adequate dietary calcium consumption is crucial for maintaining BMD and bone health, the possible CVD risks associated with excessive calcium intake should be considered in formulating calcium intake guidelines for adults [121].

A retrospective clinical study revealed that vitamin D administration alone mitigated pathological ossification, while the combination of vitamin D and calcium exacerbated pathological ossification in the majority of patients (*p* < 0.0001), irrespective of disease type and patient age. Bone mineral density (BMD) measurements showed a decreasing trend in the group with vitamin D alone, whereas an increasing trend was observed in the vitamin D and calcium combination group [122].

Calcium and vitamin D have been regarded as beneficial for bone metabolism, potentially affecting the survival of arthroplasties. The association between calcium and vitamin D use and the revision rate after primary total knee arthroplasty was also examined. The combination of calcium and vitamin D at a dose of 800 IU or greater for over one year was associated with the greatest reduction in the risks for revision surgery following knee arthroplasty [123].

Combining these dietary approaches with regular physical activity can significantly improve joint health, potentially enhancing the quality of life for those with osteoarthritis.

## 12. Pharmacological Strategies for the Treatment of OA

Hyaluronic acid (HA) injections remain a widely used approach in OA management, particularly for knee osteoarthritis. A recent meta-analysis comparing platelet-rich plasma (PRP) combined with HA versus PRP alone demonstrated that HA+PRP therapy provided superior pain relief and functional improvement [124].

Low molecular weight HA is distinguished by its anti-inflammatory properties, as it reduces the production of inflammatory mediators and modulates the immune response within the conversely high molecular weight HA functions as a shock absorber, enhancing the viscoelastic properties of synovial fluid and providing a mechanical protective effect on the joint.

The combination of high and low molecular weight hyaluronic acid (HA) in hybrid formulations, such as those utilizing NAHYCO^®^ Hybrid Technology, offers a dual benefit by simultaneously addressing both mechanical and inflammatory factors [119].

The authors of [125] reported that Hylan G-F 20, a cross-linked HA injection, provides significant pain relief lasting up to six months for patients with knee OA. However, its efficacy varies depending on disease severity, with better results in early-to-moderate OA [126].

Additionally, a survey-based study indicated that clinicians perceive HA alternatives, such as intra-articular polynucleotide (PN) injections, to be more effective than HA alone, particularly for chronic pain and joint cushioning [127].

Emerging therapies have also been explored, including amniotic suspension allografts (ASAs), which have demonstrated higher efficacy than HA in pain reduction and functional improvement [125]. Meanwhile, microfragmented adipose tissue (MAT) injections have been tested alongside HA and PRP, but a systematic review found no significant advantage for MAT over other orthobiologics [128].

## 13. Conclusions

The landscape of cartilage diagnostics and dietary intervention is rapidly evolving, driven by advances in imaging technologies, molecular biology, and bioengineering. The traditional reliance on structural imaging modalities like X-ray and MRI is being complemented by innovative approaches which offer insights into the biochemical and molecular underpinnings of cartilage health. Techniques such as sodium MRI, molecular imaging, and wearable biosensors provide the ability to detect early cartilage degeneration long before clinical symptoms appear, which is crucial for effective intervention and management of conditions like osteoarthritis. At the same time, the role of diet in managing cartilage health is gaining increasing attention, as certain nutrients have been shown to support cartilage integrity and may aid in slowing disease progression. Essential nutrients like omega-3 fatty acids, antioxidants, and vitamin D are not just important for overall health but have also been shown to help repair cartilage and reduce inflammation. By bringing together nutrition-focused strategies with advanced diagnostic tools, healthcare providers can offer a more well-rounded approach to managing cartilage-related conditions. This combination not only supports early detection and personalized treatment but also empowers patients to take charge of their health with lifestyle choices which promote long-term joint well-being.

The integration of these advanced diagnostic tools into clinical practice has the potential to transform the management of cartilage-related diseases, enabling earlier diagnosis, more precise monitoring of disease progression, and the development of personalized treatment strategiesAs these technologies continue to advance and become more widely available, they will likely enable timely and effective intervention following early diagnosis.However, to fully realize this potential, ongoing research, large-scale clinical validation, and efforts to make these tools more accessible and cost-effective are essential. The future of cartilage diagnostics lies in a multidisciplinary approach that combines cutting-edge technology with a deep understanding of cartilage biology, ultimately improving patient outcomes in cartilage-related diseases.

## Figures and Tables

**Table 1 biomedicines-13-00570-t001:** This table provides a comprehensive comparison of various imaging techniques used for diagnosing and monitoring osteoarthritis (OA). It highlights their advantages, limitations, and accuracy in evaluating cartilage integrity, including both structural and biochemical aspects. In summary, T1ρ Imaging, dGEMRIC, and gagCEST offer the highest accuracy for detecting early biochemical changes in cartilage. Techniques like T2 mapping, hybrid PET-MRI, and OCT provide high accuracy for identifying structural damage. Conventional MRI and sodium MRI offer moderate accuracy, while US has lower accuracy for assessing intra-articular changes.

Technique	Pros	Cons	Accuracy
MRI (Conventional)	Gold standard for cartilage imaging, high soft tissue contrast, well-established.	Unable to detect microscopic cartilage lesions or early degeneration.	High for cartilage morphology, low for early-stage biochemical changes.
T2 Mapping	Sensitive to water content and collagen integrity, improves early detection of cartilage degeneration.	Affected by collagen matrix disruption, requires advanced MRI protocols.	High for detecting early cartilage degeneration and water content changes.
T1ρ Imaging	Sensitive to proteoglycan content, less influenced by magic angle effect, predicts cartilage degeneration.	Technically complex, requires advanced imaging sequences, limited availability.	High for early-stage OA, sensitive to proteoglycan content.
dGEMRIC	Quantifies GAG content, predicts OA development, effective for early detection.	Requires gadolinium contrast, potential risks for kidney patients, long imaging process.	High for GAG quantification, less accurate in advanced OA.
Sodium MRI	Measures sodium ion concentration linked to GAGs, noninvasive assessment of GAGs.	Low SNR, requires high-field MRI, less effective for stiffness assessment.	Moderate for GAG content, low due to SNR issues.
gagCEST	Direct quantification of GAGs, detects early cartilage damage, high spatial resolution.	Technically demanding, lower availability, complex interpretation.	High for detecting early GAG changes, extremely precise for biochemical insights.
Hybrid PET-MRI	Combines metabolic and structural imaging, detects subchondral bone activity early.	Limited data in the literature, expensive and complex technology.	High for combining metabolic and structural data, limited data in cartilage applications.
Ultrasound	Widely available, cost-effective, reliable for detecting osteophytes, synovitis, and joint effusion.	Cannot visualize intra-articular structures well, limited to superficial abnormalities.	Moderate for superficial abnormalities, low for intra-articular changes.
Optical Coherence Tomography (OCT)	High-resolution, detects early microstructural changes in cartilage, real-time imaging.	Relatively new, limited clinical data, high technical complexity.	Extremely high for early microstructural changes, experimental in OA.

**Table 2 biomedicines-13-00570-t002:** Overview of biochemical markers for cartilage health, including collagen breakdown products, aggrecan fragments, and inflammatory markers. This table highlights their respective pros, cons, and accuracy in detecting cartilage degeneration, inflammation, and early osteoarthritis (OA) progression. Collagen breakdown products and aggrecan fragments show high sensitivity in detecting cartilage degradation, while inflammatory markers offer insights into general inflammation but are less specific to cartilage damage.

Biomarker Type	Pros	Cons	Accuracy
Collagen Breakdown Products (CTX-II, C2C)	Widely studied, reflect cartilage degradation, good correlation with OA progression.	Variability due to age, gender, and other factors, lack of standardized assays.	High accuracy for detecting collagen degradation, elevated in OA (e.g., urinary CTX-II).
Aggrecan Fragments	Early indicator of cartilage breakdown, especially sensitive in early OA stages.	May be influenced by other biological processes, less studied than collagen markers.	High sensitivity for early-stage OA, especially in detecting early proteoglycan loss.
Inflammatory Markers (CRP, IL-6)	Provide insight into inflammation contributing to cartilage degradation.	Non-specific to cartilage degradation, markers of general inflammation.	Moderate, as they indicate inflammation but not specific to cartilage damage.

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
