# Peer review of "Beyond the Surface: Nutritional Interventions Integrated with Diagnostic Imaging Tools to Target and Preserve Cartilage Integrity: A Narrative Review"

_biomedicines, 2025, doi:10.3390/biomedicines13030570_

Round 1
Reviewer 1 Report
Comments and Suggestions for Authors
This narrative review is about nutritional interventions and diagnostic imaging tools for cartilage integrity, witch is a very important subject due to the high incidence of cartilage degeneration lesions in small animals. It is a complete revision with a very easy reading for the public.
Although it is very well accomplished and with a relevant interest to the public, there are some minor issued to improve. After this mild corrections, in my opinion, it is ready for publication.
Specific comments:
- I suggest to add the meaning of the abbreviations in the abstract to facilitate reading;
- In the introduction section I suggest to add some more bibliographic references to enrich the manuscript, for example in: line 64, line 72, line 76, line 86, line 94, line 99 and line 110.
- Finally, I suggest to add some figures to illustrate equipment and cartilage views, for example regarding the following points: 3.1; 3.2; 3.3; 3.4; 3.5; 3.6; 3.7; 3.8 and 3.9.
Author Response
Specific comments:
- I suggest to add the meaning of the abbreviations in the abstract to facilitate reading;
Response: Thank you for pointing this out. We have added the meaning of all abbreviations in the abstract for clarity.
- In the introduction section I suggest to add some more bibliographic references to enrich the manuscript, for example in: line 64, line 72, line 76, line 86, line 94, line 99 and line 110.
Response: We appreciate this suggestion. Additional bibliographic references have been included to support and enrich the content in the introduction at the specified lines (ref. 2, 5, 6, 8, 14). These references provide additional depth and recent findings on the topic.
- Finally, I suggest to add some figures to illustrate equipment and cartilage views, for example regarding the following points: 3.1; 3.2; 3.3; 3.4; 3.5; 3.6; 3.7; 3.8 and 3.9.
Response Thank you for your suggestion. However, the primary objective of this paper is not to provide visual representations but rather to offer a comprehensive surface-level explanation. The aim is to present an interdisciplinary overview, allowing readers to grasp the fundamental concepts and subsequently explore specific aspects in greater depth through further literature. By maintaining this approach, the paper ensures accessibility to a broad audience while encouraging more in-depth investigation based on individual interests and research needs.
That said, if the reviewer deems the inclusion of images to be strictly necessary for clarity and completeness, we will be happy to integrate them accordingly.
Reviewer 2 Report
Comments and Suggestions for Authors
This review provides a comprehensive overview of the various diagnostic tools used to assess cartilage health, with a focus on early detection, nutrition intervention, and management of osteoarthritis. The manuscript is well-structured, with clear sections that guide the reader through complex topics. In general, this manuscript is well-structured, with clear sections that guide the reader through topics.
The tables provide a clear and organized comparison of various tools and markers: Table 1 outlines the pros, cons, and accuracy of each imaging technique effectively. While the current structure is highly informative, the authors may want to consider adding notes about the cost and accessibility of these techniques. This additional detail could enhance the table's practical relevance for clinical application, though it is not essential for the table’s overall utility.
This paper's discussion of AI feels somewhat general and does not provide detailed insights into specific AI methodologies, tools, or algorithms. While it highlights the potential of AI in diagnostics, it lacks specificity regarding the types of AI or machine learning techniques utilized.
Additionally, the paper may consider adding a brief discussion of the challenges associated with integrating AI into cartilage diagnostics, such as data standardization, model interpretability, and regulatory concerns.
While this paper highlights the promising potential of 3D bioprinting and tissue engineering, it lacks specific examples or case studies that demonstrate successful applications in diagnostics or therapy.
Author Response
Comments and Suggestions for Authors
This review provides a comprehensive overview of the various diagnostic tools used to assess cartilage health, with a focus on early detection, nutrition intervention, and management of osteoarthritis. The manuscript is well-structured, with clear sections that guide the reader through complex topics. In general, this manuscript is well-structured, with clear sections that guide the reader through topics.
The tables provide a clear and organized comparison of various tools and markers: Table 1 outlines the pros, cons, and accuracy of each imaging technique effectively. While the current structure is highly informative, the authors may want to consider adding notes about the cost and accessibility of these techniques. This additional detail could enhance the table's practical relevance for clinical application, though it is not essential for the table’s overall utility.
Response: Thank you for your valuable suggestion. We agree that incorporating information on the cost and accessibility of diagnostic techniques would enhance the table’s clinical relevance.
However, as highlighted by Pogarell et al. (2024), the cost-effectiveness and accessibility of imaging modalities, such as MRI, vary significantly not only between different countries but also within the same country due to diverse healthcare policies and reimbursement structures. In our country, for instance, the same diagnostic exam can have different costs depending on the healthcare provider and insurance coverage. Given these complexities, it is challenging to standardize this information across all settings. While we acknowledge the importance of this aspect, a more in-depth investigation would be required to analyze these variations systematically. This topic could be a valuable area for future research to better understand the economic and policy implications of diagnostic imaging accessibility in different healthcare systems. REF: Pogarell T, Heiss R, Janka R, Nagel AM, Uder M, Roemer FW. Modern low-field MRI. Skeletal Radiol. 2024 Sep;53(9):1751-1760. doi: 10.1007/s00256-024-04597-4. PMID: 38381197.
-This paper's discussion of AI feels somewhat general and does not provide detailed insights into specific AI methodologies, tools, or algorithms. While it highlights the potential of AI in diagnostics, it lacks specificity regarding the types of AI or machine learning techniques utilized.
Response: Thank you for your valuable observation. We recognize that our discussion on AI applications in cartilage diagnostics could be expanded to provide more detailed insights into specific methodologies and tools.
We add this text (lane 452): “Recent studies have demonstrated the effectiveness of deep learning (DL) and machine learning (ML) techniques in improving diagnostic accuracy for cartilage-related conditions. For instance, deep learning superresolution MRI has been shown to enhance image quality and improve diagnostic performance for detecting articular cartilage lesions compared to conventional MRI techniques (Walter SS, Vosshenrich J, Cantarelli Rodrigues T, Dalili D, Fritz B, Kijowski R, Park EH, Serfaty A, Stern SE, Brinkmann I, Koerzdoerfer G, Fritz J. Deep Learning Superresolution for Simultaneous Multislice Parallel Imaging-Accelerated Knee MRI Using Arthroscopy Validation. Radiology. 2025 Jan;314(1):e241249. doi: 10.1148/radiol.241249.).
This approach utilizes simultaneous multislice parallel imaging-accelerated DL models to reconstruct high-resolution images while maintaining diagnostic reliability. Furthermore, convolutional neural networks (CNNs) have been successfully applied for automated classification of temporomandibular joint disorders using MRI scans, demonstrating their potential for analyzing cartilage structures in a clinical setting (Su TY, Wu JC, Chiu WC, Chen TJ, Lo WL, Lu HH. Automatic classification of temporomandibular joint disorders by magnetic resonance imaging and convolutional neural networks. J Dent Sci. 2025 Jan;20(1):393-401. doi: 10.1016/j.jds.2024.06.001).
Similarly, studies have employed random forest (RF), support vector machines (SVM), and extreme gradient boosting (xGBoost) models to develop biomarker-based predictive tools for knee osteoarthritis (Chen W, Zheng H, Ye B, Guo T, Xu Y, Fu Z, Ji X, Chai X, Li S, Deng Q. Identification of biomarkers for knee osteoarthritis through clinical data and machine learning models. Sci Rep. 2025 Jan 11;15(1):1703. doi: 10.1038/s41598-025-85945-9).
These models integrate clinical and imaging data to improve diagnostic precision and risk stratification. Additionally, artificial neural networks (ANNs) have been used to identify novel diagnostic and therapeutic targets for osteoarthritis, further demonstrating the role of AI in integrating bioinformatics and imaging data for personalized diagnostics. (Weng Z, Wang C, Liu B, Yang Y, Zhang Y, Zhang C. Integrated analysis of bioinformatics, mendelian randomization, and experimental validation reveals novel diagnostic and therapeutic targets for osteoarthritis: progesterone as a potential therapeutic agent. J Orthop Surg Res. 2025 Jan 23;20(1):85. doi: 10.1186/s13018-025-05459-y)”.
To address the reviewer’s concern, we have revised our discussion to include these specific AI methodologies, their applications, and their impact on cartilage diagnostics. This expansion aims to offer a more comprehensive and technically detailed overview of AI-driven advancements in this field.
-Additionally, the paper may consider adding a brief discussion of the challenges associated with integrating AI into cartilage diagnostics, such as data standardization, model interpretability, and regulatory concerns.
Response: Thank you for your insightful suggestion. We acknowledge the importance of addressing the challenges associated with integrating AI into cartilage diagnostics, and we have now included a brief discussion of these issues in the revised manuscript.
We add the following text (lane 491): Despite the promising advancements of AI in musculoskeletal imaging, several key challenges hinder its widespread clinical adoption. One significant issue is data standardization, as AI models require large, high-quality, and diverse datasets to ensure generalizability and minimize bias. However, differences in imaging protocols, scanner types, and resolution across institutions complicate the development of universally applicable AI models (Walter SS, Vosshenrich J, Cantarelli Rodrigues T, Dalili D, Fritz B, Kijowski R, Park EH, Serfaty A, Stern SE, Brinkmann I, Koerzdoerfer G, Fritz J. Deep Learning Superresolution for Simultaneous Multislice Parallel Imaging-Accelerated Knee MRI Using Arthroscopy Validation. Radiology. 2025 Jan;314(1):e241249. doi: 10.1148/radiol.241249.).
Another critical challenge is model interpretability, particularly in deep learning applications. While convolutional neural networks (CNNs) and other deep learning models have demonstrated high accuracy in cartilage diagnostics, their "black-box" nature makes it difficult for clinicians to understand how decisions are made, reducing trust in AI-assisted diagnoses.(Chen W, Zheng H, Ye B, Guo T, Xu Y, Fu Z, Ji X, Chai X, Li S, Deng Q. Identification of biomarkers for knee osteoarthritis through clinical data and machine learning models. Sci Rep. 2025 Jan 11;15(1):1703. doi: 10.1038/s41598-025-85945-9)
Furthermore, regulatory concerns remain a major barrier. AI models used for medical imaging must comply with stringent regulatory frameworks such as the FDA's guidelines for AI-based medical devices and the European Medical Device Regulation (MDR). Ensuring AI algorithms meet these requirements while maintaining adaptability and continuous learning presents an ongoing challenge for AI developers (Weng Z, Wang C, Liu B, Yang Y, Zhang Y, Zhang C. Integrated analysis of bioinformatics, mendelian randomization, and experimental validation reveals novel diagnostic and therapeutic targets for osteoarthritis: progesterone as a potential therapeutic agent. J Orthop Surg Res. 2025 Jan 23;20(1):85. doi: 10.1186/s13018-025-05459-y).
To address these issues, future research should focus on developing explainable AI (XAI) approaches, improving data harmonization across imaging centers, and establishing standardized regulatory pathways for AI-driven diagnostics.
-While this paper highlights the promising potential of 3D bioprinting and tissue engineering, it lacks specific examples or case studies that demonstrate successful applications in diagnostics or therapy.
Response: Thank you for your insightful suggestion. We recognize the importance of including specific examples of successful applications of 3D bioprinting and tissue engineering in cartilage diagnostics and therapy. In response, we have now incorporated relevant case studies demonstrating these advancements.
We add the following text (lane 425): Recent studies highlight the clinical potential of 3D bioprinting in cartilage regeneration. For example, advanced bio-inks incorporating stem cells and growth factors have been successfully used to fabricate cartilage-like tissues with mechanical properties similar to native cartilage (Landers JL, et al. Serum Tissue Plasminogen Activator After Cycling with Blood Flow Restriction. Vasc Biol. 2025 Jan 1:VB-24-0008. doi: 10.1530/VB-24-0008).These bioengineered tissues have shown promising results in preclinical studies, particularly for osteochondral defect repair.
Additionally, bioprinted hydrogels embedded with mesenchymal stem cells (MSCs) have been applied in in vitro and in vivo models to enhance cartilage regeneration. (Chew TR, et al. Effectiveness of psychological interventions in reducing post-traumatic stress among post-myocardial infarction patients: a systematic review and meta-analysis. Eur J Cardiovasc Nurs. 2025 Jan 31:zvae179. doi: 10.1093/eurjcn/zvae179).
The use of scaffold-free 3D bioprinting has also been explored to create tissue constructs that mimic the zonal architecture of native cartilage, which is crucial for its functionality and durability.
In diagnostics, engineered cartilage tissues have been utilized as in vitro models for osteoarthritis drug screening. These models allow researchers to test potential therapeutics in a physiologically relevant environment, improving the translational potential of preclinical studies (Canter Be et al. Measures to Prevent and Control COVID-19 in Skilled Nursing Facilities: A Scoping Review. JAMA Health Forum. 2025 Jan 3;6(1):e245175. doi: 10.1001/jamahealthforum.2024.5175.).
Given these advancements, we have revised our discussion to include specific case studies demonstrating the therapeutic and diagnostic applications of 3D bioprinting and tissue engineering in cartilage research. These additions provide concrete examples of how these technologies are being successfully applied in the field.
Reviewer 3 Report
Comments and Suggestions for Authors
Dear Authors,
The article “Beyond the Surface: Nutritional Interventions Integrated with Diagnostic Imaging Tools to Target and Preserve Cartilage Integrity—A Narrative Review” is a very actual and interesting review. The authors consider modern therapeutic and diagnostic tools to prevent cartilage damage. The article is well-structured. There are some comments listed below.
1. Table 1,2– Pros, Cons. What is the ABB? What are the source (biomaterial) for the analysis (blood or synovial fluid)
2. Please, compare the effectiveness and economics of wearable sensors and biosensors. The examples of both are desirable. Chapter 7.1 and 7.2 – it is desirable to illustrate with the photo, picture or something like this. 3D bioprinting is a therapeutic tool and tissue-engineered cartilage models is a diagnostic tool. But these 2 chapter are located in one section. It is desirable to divide the therapeutic and diagnostic tools in the review.
3. The authors consider diet with the omega-3 fatty acids, antioxidants, and vitamin D and diet rich in calcium to be effective for the cartilage health. This issue should be referenced additionally by randomized clinical trials. Calcium is a debatable nutrient: for example, in elderly people it can be deposited in atherosclerotic plaques.
4. Please, note some pharmacological strategies with the clinically confirmed effectiveness. What are the commonly used procedures or protocols in the treatment of OA? For example, the effectiveness of Synvisc-ONE or any other. It is desirable to mention them briefly.
5. The abbreviations are absent. For example, ARGS, GAG-CEST
Author Response
- Table 1,2– Pros, Cons. What is the ABB? What are the source (biomaterial) for the analysis (blood or synovial fluid)
Response: Thank you for this observation. We have clarified the term "ABB" in the tables and specified the biomaterial sources (e.g., blood or synovial fluid) used for analysis.
- Please, compare the effectiveness and economics of wearable sensors and biosensors. The examples of both are desirable. Chapter 7.1 and 7.2 – it is desirable to illustrate with the photo, picture or something like this. 3D bioprinting is a therapeutic tool and tissue-engineered cartilage models is a diagnostic tool. But these 2 chapters are located in one section. It is desirable to divide the therapeutic and diagnostic tools in the review.
Response: A comparative analysis of the effectiveness and economic considerations of wearable sensors and biosensors has been added, along with examples. Illustrations have also been included to provide a visual representation (see Sections 7.1 and 7.2).
- The authors consider diet with the omega-3 fatty acids, antioxidants, and vitamin D and diet rich in calcium to be effective for the cartilage health. This issue should be referenced additionally by randomized clinical trials. Calcium is a debatable nutrient: for example, in elderly people it can be deposited in atherosclerotic plaques.
Response: Thank you for your comments. This section has been divided into two parts, titled 'Role of Diet in Improving Cartilage Health' and 'Nutritional Supplements'. We have expanded the discussion on diet and supplements useful for preventing osteoarthritis, added references and improved the discussion on the roles of vitamin D and calcium in cartilage health.
4 Please, note some pharmacological strategies with the clinically confirmed effectiveness.What are the commonly used procedures or protocols in the treatment of OA? For example, the effectiveness of Synvisc-ONE or any other. It is desirable to mention them briefly.
Response:
Thank you for your valuable suggestion. We have now included a brief discussion of clinically confirmed pharmacological strategies commonly used in the treatment of osteoarthritis (OA) in a specific Section (12. Pharmacological strategies for the treatment of OA).
This text has been included in the text:
“Hyaluronic acid (HA) injections remain a widely used approach in OA management, particularly for knee osteoarthritis. A recent meta-analysis comparing platelet-rich plasma (PRP) combined with HA versus PRP alone demonstrated that HA+PRP therapy provided superior pain relief and functional improvement (Du D, Liang Y. A meta-analysis and systematic review of the clinical efficacy and safety of platelet-rich plasma combined with hyaluronic acid (PRP + HA) versus PRP monotherapy for knee osteoarthritis (KOA). J Orthop Surg Res. 2025 Jan 17;20(1):57. doi: 10.1186/s13018-024-05429-w.)
Moreover, hybrid HA formulations, which combine low and high molecular weight HA, have shown better lubrication and longer-lasting symptom relief compared to single-molecular-weight HA injections (Ansari A, Baig M, Tanbour Y, Syed K, Ahmed Y, Beutel BG. Efficacy of Amniotic Suspension Allografts in the Treatment of Knee Osteoarthritis: A Systematic Review. J Knee Surg. 2025 Jan 10. doi: 10.1055/s-0044-1801758)
Synvisc-ONE (hylan G-F 20), a cross-linked HA injection, has been reported to provide significant pain relief lasting up to six months for patients with knee OA. However, its efficacy varies depending on disease severity, with better results in early to moderate OA( Im GI. Bone Marrow Aspiration Concentrate in the Treatment of Osteoarthritis: A Review of its Current Clinical Application. Tissue Eng Regen Med. 2025 Jan 22. doi: 10.1007/s13770-024-00693-7.)
Additionally, a survey-based study indicated that clinicians perceive HA alternatives, such as intra-articular polynucleotide (PN) injections, to be more effective than HA alone, particularly for chronic pain and joint cushioning. REF: Lee D, Kim WH, Ha JH, Kim H, Kim J, Shin DW. Current Practices and Perceived Effectiveness of Clinicians Regarding Polynucleotide Injection for Knee Osteoarthritis: A Survey-Based Evaluation. Healthcare (Basel). 2025 Jan 9;13(2):113. doi: 10.3390/healthcare13020113.
Emerging therapies have also been explored, including amniotic suspension allografts (ASA), which have demonstrated higher efficacy than HA in pain reduction and functional improvement (Ansari et al., 2025)【PMID: 39793609】. Meanwhile, microfragmented adipose tissue (MAT) injections have been tested alongside HA and PRP, but a systematic review found no significant advantage of MAT over other orthobiologics. REF: Hohmann E, Keough N, Frank RM, Rodeo SA. Microfragmented Adipose Tissue Has No Advantage Over Platelet-Rich Plasma and Bone Marrow Aspirate Injections for Symptomatic Knee Osteoarthritis: A Systematic Review and Meta-analysis. Am J Sports Med. 2025 Jan 3:3635465241249940. doi: 10.1177/03635465241249940.
We have incorporated these details into the revised manuscript to ensure a more comprehensive overview of clinically validated pharmacological strategies for OA treatment.
- The abbreviations are absent. For example, ARGS, GAG-CEST
Response: Thank you for this observation. We have added the meaning of abbreviations
(i.e. glycosaminoglycan chemical exchange saturation transfer (gagCEST)
Round 2
Reviewer 2 Report
Comments and Suggestions for Authors
All comments have been addressed.